# Brain-Biomarker Changes in Body Fluids of Patients with Parkinson’s Disease

**DOI:** 10.3390/ijms241310932

**Published:** 2023-06-30

**Authors:** Cristina Cocco, Antonio Luigi Manai, Elias Manca, Barbara Noli

**Affiliations:** Department of Biomedical Sciences, University of Cagliari, 09042 Monserrato, Italy; a.manai2@studenti.unica.it (A.L.M.); elias.manca@ki.se (E.M.); barbaranoli@yahoo.it (B.N.)

**Keywords:** early diagnosis, Parkinson’s disease, VGF

## Abstract

Parkinson’s disease (PD) is an incurable neurodegenerative disease that is rarely diagnosed at an early stage. Although the understanding of PD-related mechanisms has greatly improved over the last decade, the diagnosis of PD is still based on neurological examination through the identification of motor symptoms, including bradykinesia, rigidity, postural instability, and resting tremor. The early phase of PD is characterized by subtle symptoms with a misdiagnosis rate of approximately 16–20%. The difficulty in recognizing early PD has implications for the potential use of novel therapeutic approaches. For this reason, it is important to discover PD brain biomarkers that can indicate early dopaminergic dysfunction through their changes in body fluids, such as saliva, urine, blood, or cerebrospinal fluid (CSF). For the CFS-based test, the invasiveness of sampling is a major limitation, whereas the other body fluids are easier to obtain and could also allow population screening. Following the identification of the crucial role of alpha-synuclein (α-syn) in the pathology of PD, a very large number of studies have summarized its changes in body fluids. However, methodological problems have led to the poor diagnostic/prognostic value of this protein and alternative biomarkers are currently being investigated. The aim of this paper is therefore to summarize studies on protein biomarkers that are alternatives to α-syn, particularly those that change in nigrostriatal areas and in biofluids, with a focus on blood, and, eventually, saliva and urine.

## 1. Introduction

### 1.1. PD

PD is a brain disorder with an irreversible and continuous process of neurodegeneration until death. There is no cure for this disease, but treatments are available, of which levodopa (L-dopa) is the gold standard. The disease is diagnosed by neurologists by observing motor symptoms such as bradykinesia, rigidity, postural instability, and resting tremor [1,2]. However, all these symptoms occur years (possibly decades) after the onset of the neurodegenerative process, leading to difficulties in recognizing early PD, which is characterized by hyposmia, depression, constipation, sleep disturbances (REM-sleep-behavior disorder, excessive daytime sleepiness) [3,4]. As a result, misdiagnosis rates are in the range of 10–30% [5,6,7], which reduces the therapeutic benefit and compromises the possibility of using novel therapeutic approaches. For this reason, significant effort is underway in the search for candidate biomarkers that can track disease progression and/or differentiate PD from the other neurodegenerative diseases, as well as being able to detect early changes at the level of the basal ganglia. The basal ganglia are a group of subcortical nuclei including the striatum (STR), which is subdivided into dorsal (caudate and putamen) and ventral (nucleus accumbens), the globus pallidus (external and internal parts), the subthalamic nucleus, and the substantia nigra (SN) [8]. The SN is divided into two areas: the pars compacta (SNpc), which consists mainly of dopaminergic perikarya, and the pars reticulata (SNpr), which consists mainly of gamma-aminobutyric acid (GABA)-inhibitory neurons. The SNpc has dopaminergic projections to the striatum, putamen, and caudate nuclei. Within the striatum, these dopaminergic projections synapse on the D1- and D2-family receptor neurons, which are involved in the “direct and indirect pathways” of the basal ganglia [9]. Furthermore, PD is not only characterized by alterations in the dopaminergic system, but the map of altered substances includes several neuroproteins, neurotransmitters, and neuropeptides that are expressed or not expressed within the nigro-striatal circuits [10,11,12].

### 1.2. PD Biomarkers: General Information

Because PD is a multifactorial disease, several types of biomarkers are under study, including clinical, brain-based (such as neuroimaging), genetic, and biochemical. Clinical biomarkers, such as bradykinesia, resting tremor, and muscle rigidity are used by clinicians to identify PD and monitor both response to medical treatment and disease progression. However, other clinical features, such as hyposmia, rapid eye movement (REM) sleep-behavior disorder (RBD), and constipation are under consideration as biomarkers of prodromal PD [3,4]. Neuroimaging techniques, such as transcranial sonography, magnetic resonance imaging (MRI), positron-emission tomography (PET) or single-photon-emission computed tomography (SPECT) are considered biomarkers of nigrostriatal neurodegeneration and may also detect the early phase of PD and follow its progression [13]. Genetic biomarkers are considered important tools in the diagnosis of PD, and a family history is reported in approximately 10–20% of PD patients [14]. Of the various genes identified as biomarkers, some are associated with autosomal recessive inheritance, particularly Parkin and PINK1, which are associated with typical early-onset PD [15]; others are related to atypical juvenile PD, such as ATP13A2, DNAJC6, FBOX7, and SYNJ1 [16], whereas SNCA, LRRK2, and VPS35 are associated with typical autosomal dominant PD [17]. In general, studies of genetic risk factors for PD have been limited to single-nucleotide variants, which account for a small fraction of the genetic variation in the human genome. Recent results from researchers involved in the PD Biomarker Programme (PDBP: https://pdbp.ninds.nih.gov/news, accessed on 6 June 2023) identified a structural-genotype variant associated with PD risk in 2585 patients compared with 2779 controls [18]. In terms of biochemical biomarkers, there are a large number of studies focusing on different biomolecules present in the brain and/or body fluids, including metabolites as glutathione, pyruvate, and glycine derivatives [19], plasma lipids such as N-acylphosphatidyl ethanolamines [20], and a large number of proteins, including α-syn. There are several methods used to evaluate biochemical biomarkers. Quantitative reverse-transcription PCR (RT-qPCR) or in situ hybridization can be used to identify the mRNA, while electronic, fluorescence and confocal microscopy may be useful to investigate where a molecule is expressed and/or in which cell types. However, the most useful techniques for detecting biomarkers in the blood include multicolor flow cytometry, the radioimmunoassay (RIA), the enzyme-linked immunosorbent assay (ELISA), the single-molecule array (Simoa), and chemiluminescence. In the last decade, high-performance liquid chromatography–mass spectrometry (HPLC-MS) has become an essential part of biomarker-discovery research through targeted, semi-targeted or non-targeted measurement analysis, allowing large amounts of data to be collected with high accuracy from a variety of samples, including bodily fluids. Interestingly, through a spectrometry-based method, the analysis of SN from PD patients and controls in a PDBP study allowed the identification of several mechanisms leading to dopaminergic cell death and α-syn deposition, including those related to ribosome activity and GABAergic synapse [21]. Following the identification of the crucial role of brain α-syn in the pathology of PD, a very large number of studies summarized the changes to this protein in body fluids, particularly in the CSF. However, methodological problems have led to the poor diagnostic/prognostic value of this protein, as shown in several studies [22,23,24], including one carried out in a large Parkinson’s Progression Markers Initiative (PPMI) cohort of patients [25], and alternative biomarkers are currently under investigation. The aim of this paper is, therefore, to focus on brain-protein biomarkers, alternatives to α-syn and, in particular, those that may indicate early dopaminergic dysfunction through their changes in bodily fluids. With regard to CSF, despite its close association with the brain, the invasiveness of sampling is a limitation, whereas the other bodily fluids are easier to obtain and could also allow population screening, so the present review focuses on blood and, eventually, saliva and urine. To the author’s knowledge, reviews on brain-protein biomarkers that change in the biofluids are rare. These candidate biomarkers are reviewed for their potential to aid in early diagnosis, prognosis, or monitoring of specific clinical features.

## 2. Brain-Biomarker Changes in Body Fluids of PD Patients

### 2.1. Neurofilament Light Protein

Neurofilaments (NFs) are members of a family of intermediate filament proteins characterized by an ‘intermediate’ diameter (10 nm), which is larger than that of actin filaments (6 nm) and smaller than that of myosin filaments (15 nm) [26], and include NF-L, -M, and -H, corresponding to light, medium, and heavy, respectively [27]. These NF proteins occupy the axonal cytoplasm, support neurons, and interact with several other proteins that regulate synaptogenesis and neurotransmission [28,29]. The -L, -M, and -H NFs were found by immunohistochemistry (IHC) to be abnormally aggregated within Lewy bodies in the post-mortem SN of a PD patient (whose disease stage was not reported) [30]. Blood NF-L levels measured by ELISA were used to discriminate PD from atypical parkinsonism [31] with the same diagnostic accuracy as CSF NF-L [32]. In addition, plasma NF-L was correlated with disease severity and the progression of both motor and cognitive functions using an electrochemiluminescence immunoassay [33], while when the serum NF-L was measured by Simoa together with other fluid biomarkers, it helped to discriminate PD patients from controls [34]. In a complete longitudinal PPMI cohort study, a large dataset of NF-L measurements by Simoa or ELISA was obtained from: (i) 176 CSF samples, including newly diagnosed and drug-naïve PD patients and controls, (ii) 1190 sera from patients with PD (including newly diagnosed and drug-naïve patients) (iii) other neurodegenerative diseases, (iv) subjects with prodromal conditions and mutation carriers as well as sera form healthy controls, [35]. The results showed that the serum and CSF NF-L levels were significantly correlated and were highest in other neurodegenerative diseases, but higher in the PD patients than in the controls. Furthermore, the NF-L levels increased over time and with age (by 3.35% per year of age, and women had a median serum level 6.79%, which was higher than that of the men), were correlated with PD severity and, most importantly, were highest in early PD, but also relatively high in the prodromal groups, including patients affected by RBD disorders, suggesting a prognostic role.

Similar results were obtained in another large study [36], in which PD patients were selected on the basis of diagnosis within 2 years, not receiving treatment at baseline, Hoehn and Yahr (H-Y) stage 1 or 2, 123I-isoflurane DaT imaging, and, to avoid misdiagnosis, a longitudinal review of their diagnosis. The study included CSF samples from 207 PD patients and 102 controls, while the sera were collected from 361 PD patients and 176 controls, plus 291 individuals who provided both serum and CSF samples. Using Simoa, the baseline CSF and serum NF-L levels were both higher in the PD patients than in the controls, but the CSF NF-L levels were significantly elevated in the males compared to the females, which is not consistent with the study cited above. In addition, the same study showed in a cross-sectional and prospective follow-up of the PD patients that baseline CSF and serum NF-L levels could predict motor decline and tremor (but not rigidity). Interestingly serum NF-L levels measured by Simoa may also be useful to differentiate PD from essential tremor (ET) [37]. Indeed, when NF-L levels were measured by electrochemiluminescence immunoassay in sera from patients with PD (>2 years of diagnosis; *n* = 146), ET (*n* = 82) and 60 age-matched healthy controls, significantly higher levels were found in PD than in ET and healthy controls. In the same study, serum NF-L was found to be elevated in patients with advanced H-Y stage and dementia and was an independent contributor to motor symptoms and cognitive severity. In 289 patients with advanced PD, NF-L measured by Simoa was associated with motor function, cognitive decline, and subclinical cardiac damage [38]. A study focusing on the comparability of PPMI and Asian cohorts highlighted the importance of adjusting several variables, including demographics, age, and sex, to best interpret serum NF-L levels for research and clinical practice [39]. However, despite all these important findings, the diagnostic accuracy of serum NF-L is low [35] and changes are not specific for PD [40,41,42,43]. Therefore, rather than being a diagnostic tool for PD, NF-L may be a useful marker for certain specific aspects of PD, such as dementia, cognitive decline, or cardiac damage. It may also be a good biomarker for distinguishing PD from ET.

### 2.2. Substance P

Substance P (Sub-P) is an 11-amino-acid peptide that belongs to the tachykinin family and is derived from a pre-protachykinin polyprotein precursor of the TAC1 gene [44,45]. Sub-P acts through various mechanisms, including autocrine, paracrine, and endocrine, to modulate pain perception by altering cellular signaling pathways [46,47]. Sub-P mRNA is found in the GABAergic neuron terminals of the SN [48], where the protein acts to sensitize postsynaptic dopaminergic neurons to glutamate [49]. In humans, the presence of Sub-P within α-syn aggregates was demonstrated by IHC in the perikarya of the colon [50] and olfactory bulb [51]. Sub-P changes in SN and/or STR of PD patients have been extensively studied in both animal models and patients. Sub-P levels detected by IHC and/or RIA were decreased in the SN and STR of both unilateral and bilateral rats injected with 6-hydroxydopamine (6-OHDA) [52,53,54], whereas the opposite results were obtained in rats treated with intraperitoneal administration of 1-methyl-4-phenyl-1,2,3,6-tetrahydropyridine (MPTP) [55,56]. Consistent with animal models, when human post-mortem brains were measured using various techniques, including RIA, reduced levels of Sub-P were found in the SN [57,58,59], globus pallidus [59], putamen [58,60], and caudate [61] of PD patients compared with control brain samples. In patients with early (*n* = 15) and advanced (*n* = 15) PD, sputum Sub-P levels were assessed by ELISA and were significantly decreased in the advanced-stage patients compared to the controls and early-stage patients [62]. In another study [63] including 22 PD patients (with a score of 2 on the H-Y scale and treated with L-dopa) and 12 controls, serum Sub-P levels were measured by competitive ELISA. The Sub-P levels were higher than in the controls and increased proportionally with the disease severity (motor impairment). The limited number of available studies on Sub-P in the bodily fluids of PD patients and the small number of subjects involved make it difficult to identify Sub-P as a reliable blood (or saliva) biomarker. However, the presence of Sub-P within α-syn aggregates needs to be investigated further, especially for its possible involvement in early PD.

### 2.3. S100 Calcium-Binding Protein A10

The S100s are a family of low-molecular-weight proteins with two calcium-binding sites that have a helix–loop–helix conformation. Approximately 21 different proteins have been recognized and are encoded by the S100A1, S100A2, and S100A3 genes (among others) genes [64]. The S100 calcium-binding protein A10 (namely S100A10 or P11), encoded by the S100A10 gene, modulates neuronal function [65]. The expression of P11 was investigated in post-mortem brains (both the SN and the putamen from fresh snap-frozen samples) of PD patients (3 females and 2 males) and controls (4 females and 1 male). The P11 levels, which were submitted to a Western blot (WB) analysis, decreased in the dopaminergic SN cells, along with the mRNA detected by laser-capture microdissection coupled to quantitative real-time PCR [66]. In the same study, P11 was also investigated by multicolor flow cytometry in leukocytes from a total of 42 PD patients with or without depression (at different disease stages, including de novo PD patients and those with H-Y scores of) and 15 controls. The P11 levels in the CD8+ cells were found to be higher in the depressed and non-depressed PD patients compared to the controls (with a sensitivity and specificity of 93%), but increased exclusively in the CD14+CD16− of the depressed PD patients, corresponding to the same modulation reported in depressed patients without PD [67]. Furthermore, the P11 levels in the monocytes, cytotoxic T cells, and NK cells were all correlated with PD severity. Although the pathological role of P11 in PD remains unclear, as with its role in normal SN, P11 can be considered as a PD biomarker that is useful in identifying patients at risk of developing depression [68].

### 2.4. Neurotensin

Neurotensin (NT) is a 13-amino-acid peptide [69] that is widely distributed in neurons, in which, in dense nuclear vesicles, it acts as a neuromodulator of several neurotransmitters, such as acetylcholine, as well as serotonin, GABA, and dopamine [70].The IHC of normal rat brains showed the expression of NT in ventral tegmental-area fibers and SN (both compacta and reticulata) [71], while combined HPLC/RIA showed an increased (twofold) expression in post-mortem SN samples from 6 advanced-PD patients (compared to 5 controls) [72]. In another study, using specific RIA on postmortem brain samples from 25 human patients with advanced PD compared to controls, the NT levels were significantly decreased in the hippocampus but not altered in the SN [73]. Using RIA on blood samples from 36 subjects (16 controls,16 patients with advanced disease, and 4 untreated patients), the NT concentrations were higher in the PD patients than in the controls and higher in the untreated patients than in the patients treated with L-dopa [74]. However, the number of studies using blood from early PD patients is excessively small, so further studies are needed to clarify the role of NT as a blood biomarker in PD.

### 2.5. Chromogranins

Chromogranin proteins [75] are the major components of secretory granules in neuroendocrine cells, where they play a role in granulogenesis and in the sorting and processing of secretory-protein cargo [76]. Chromogranin proteins are considered biomarkers of neuroendocrine neoplasia [77]. Many proteins belong to the chromogranin family, including chromogranins A (CgA) and B (CgB) and secretogranin II (SgII, or chromogranin C) [78]. Several studies have reported that chromogranin-family proteins are found in the SN and are involved in PD changes. One study reported that CgA, detected by IHC with an antibody against the WE14 epitope, was expressed in non-dopaminergic neurons of the SNpc [79]. Processed and unprocessed forms of CgA were expressed in anatomically defined GABAergic, glutamatergic, cholinergic, and catecholaminergic neurons in the rodent central nervous system [79]. In human studies, the midbrain, cerebellum, and cerebrum of 7 PD patients and 2 PD-associated Alzheimer’s disease (AD) patients were examined by light and electron microscopy using antibodies against synaptophysin and ChA, and compared with 10 controls. Lewy bodies were observed in the hematoxylin- and eosin-stained sections of all the PD and PD/AD cases. Within the Lewy bodies, the antibodies mainly stained the peripheral zones, and at the ultrastructural level, labeling was found in the vesicular, filamentous, and granular structures. However, the antibodies did not show any differences between the PD and normal controls on immunoblot analysis [80]. In another study, post-mortem brains from 4 cases of AD, 2 cases of Pick’s disease, 4 cases of PD and various other diseases, and brains from 6 cases without neurological disorders were examined for CgA expression by IHC. Lewy bodies with vesicles in the periphery of the central nucleus were immunoreactive for CgA in the SN of the PD brain samples but also in those with other diseases [81]. Another study [82] showed that serum levels of chromogranins and secretogranins were correlated with the progression and severity of the PD. The subjects included in the study were patients with early (*n* = 14), intermediate (*n* = 18), or late (*n* = 4) stages of PD, according to their H-Y scores, as well as well-defined PD (*n* = 36) and healthy controls (*n* = 52). The serum concentrations of CgA, CgB, and SgII were measured by ELISA. Compared with the controls, the serum CgA levels were significantly increased and the serum SgII levels were significantly decreased in the overall population of PD patients, whereas the serum CgB levels did not differ between the two groups. In the early-stage group, the CgA and SgII levels were lower and higher, respectively, than in the other groups. Both the CgA and the SgII serum levels changed progressively over time and were correlated with both the H-Y and the UPDRS scores. The results for the chromogranins were promising, but the number of early-stage PD patients studied was small, and it remains to be investigated whether the blood changes are characteristic of PD or of neurodegenerative diseases in general, as has been observed using CSF [83].

### 2.6. VGF

The VGF gene (not abbreviated) is regulated by the nerve-growth factor (NGF) in PC12 cells and cultured cortical neurons [84]. The VGF precursor (proVGF), consisting of 617/615 aa in rats/mice and humans, respectively, with >85% identity (minor sequence differences between rat and human) [85], is expressed in rat brains [86], but also in human plasma [87,88]. It is induced by growth factors [89] and can give rise to a variety of truncated peptides, including the so-called TLQP [90,91,92], NERP1, and NERP2 [93]. Peptide sequences of VGF have mainly been identified in CSF, where specific peptide alterations have been associated with certain diseases. For example, VGF peptides cleaved at the N-terminus of proVGF were found to be altered in CSF of psychiatric disorders [94,95,96,97], while others in the middle of proVGF may be useful CSF biomarkers for dementia [98,99,100]. The VGF immunostaining (using an antibody against the last nine amino acids at the C-terminus) was decreased in GABAergic (containing Sub-P) fibers of the SN in rats treated with 6-OHDA, but it was restored by L-dopa treatment [101]. Unfortunately, VGF expression in human SN remains to be investigated, but VGF alterations of certain peptides (TPGH and NERP-1) have been found in the post-mortem parietal cortexes of PD patients [102]. In addition, the next-generation RNA sequencing of cingulate cortex samples (from 23 PD patients, compared with 11 controls) found the VGF gene to be implicated in the molecular etiology of PD and PD-related dementia [103]. The levels of VGF C-terminal (C-t) peptides were analyzed by a home-made competitive ELISA using an antibody directed against the last nine amino acids at the C-terminus of human proVGF in PD patients at the time of diagnosis (drug-free, *n* = 23) or after dopamine replacement (*n* = 40) compared to age-matched controls (*n* = 21) [101]. In the drug-free patients, a strong decrease (>50%) was observed at the time of diagnosis, whereas long-term L-dopa treatment caused an increase in VGF. The levels of VGF C-t peptides were also correlated with disease duration, LED, and the severity of olfactory dysfunction, but not with the H-Y score. The VGF changes observed in the blood of PD patients were verified in other bodily fluids, such as the CSF and urine. When CSF samples were analyzed from two independent cohorts of subjects (*n* = 196) and a longitudinal cohort (*n* = 105), including all the controls and treated PD patients (with disease duration of no more than 6 years), the C-terminal region of the VGF protein was found to be decreased by liquid chromatography–tandem mass spectrometry in the data-independent acquisition mode [104]. Using the same technique, more than 200 urine samples from different groups of subjects were analyzed, including (i) healthy controls, (ii) non-manifest carriers of the LRRK2 G2019S mutation, (iii) idiopathic PD patients, and (iv) manifest PD patients with LRRK2 G2019S. The results showed that changes (decrease) were found in the peptides covering most of the VGF sequence [105]. Interestingly, CSF and urine data were found to be correlated [106]. Decreased levels of VGF C-t peptides have also been studied in plasma samples from amyotrophic lateral sclerosis patients, but a small reduction was found only in the late stage of the disease [87] while no changes in plasma-VGF C-t peptides were observed in a rat model of schizophrenia [107]. In conclusion, VGF C-t peptides are potential blood biomarkers for PD, as they appear to be selectively altered in PD. However, both the use of a more specific sandwich ELISA and confirmation by HPLC-MS are required to accurately identify individual VGF sequences that are altered in PD blood.

### 2.7. Glial Fibrillary Acidic Protein

Glial fibrillary acidic protein (GFAP) is an intermediate filament protein found primarily in mature astrocytes. This protein has a much broader function than the provision of mechanical support to cells, as it is involved in the production and regulation of the blood–brain barrier, increases synaptic plasticity, and coordinates neuronal activity [108]. This protein is upregulated in brain damage or neuronal degeneration; therefore, it is considered a marker of astrocyte activation (or astrogliosis) [109]. Increased numbers of GFAP-immunoreactive astrocytes were observed in the SN of mice after only one day of MPTP administration [110], as well as after 6-OHDA treatment [111]. The role of astrogliosis in the PD brain is not fully elucidated, although most of the relevant studies observed slight or no increases in the expression of both GFAP-immunoreactive cells (determined by IHC-stereological quantification and WB analysis) in postmortem samples from SN collected from patients with PD [112,113,114]. Only Damier et al. observed the opposite results in their study, i.e., the increased expression (according to their IHC analysis) of GFAP in the SN of PD patients correlated with the percentage of lost dopaminergic neurons [115]. Changes in GFAP levels were also studied in the plasma of PD patients. Tang et al. observed an increase in GFAP levels (using Simoa) in PD patients (*n* = 60) compared to healthy controls (*n* = 15), and the highest increase was in PD patients with dementia (*n* = 24) [116]. Similar results were obtained with serum samples in which an increase in GFAP levels (detected by sandwich ELISA) was observed in PD patients (*n* = 23) compared with healthy controls (*n* = 29) [117] and, again, this increase was more pronounced in PD patients with mild cognitive impairment or dementia than in PD patients with normal cognition [118]. Another cross-sectional study examined PD patients characterized by RBD, an important risk factor for cognitive impairment in PD patients [119]. These patients (*n* = 39) had higher serum GFAP levels compared to PD patients without RBD (*n* = 70) and healthy controls (*n* = 37, measured with Simoa) [120], suggesting the role of GFAP as a predictive PD biomarker. Elevated serum levels of this protein are also observed in patients with other neurodegenerative diseases, such as AD and frontotemporal lobar degeneration [121,122].

## 3. Conclusions

PD is one of the most significant neurodegenerative diseases, and because most of the cardinal symptoms occur late in its course, a large number of dopaminergic neurons are already damaged when it is diagnosed. Therefore, there is a great need for biomarkers that can be used to ensure early diagnoses, as well as to differentiate PD from other diseases with high sensitivity and specificity, and to track disease progression. When biomarkers are altered in both the brain and body fluids it is more likely that changes in the periphery are linked to the changes occurring in the central nervous system. In addition, improvements in the identification of the neuronal proteins that change in PD could lead to a better understanding of the PD pathological mechanisms, which in turn could lead to new approaches to diagnosis and treatment. Although the candidate biomarkers studied here are promising (Table 1 and Figure 1), they have limitations related to (i) the small number of blood samples, especially in the early PD, (ii) their low specificity in PD, (iii) the lack of studies based on orthogonal methods, which are often required to provide the independent confirmation of the results. At this point, it is important to note that a single biomarker is unlikely to achieve the higher standard of sensitivity and specificity required for an accurate diagnosis of PD, as the mechanisms involved in PD are very complex and sometimes similar to those in other neurodegenerative diseases. Therefore, another good strategy may be to validate a large panel of PD fluid biomarkers, including those non-specific that may increase in value when combined. 

## Figures and Tables

**Figure 1 ijms-24-10932-f001:**
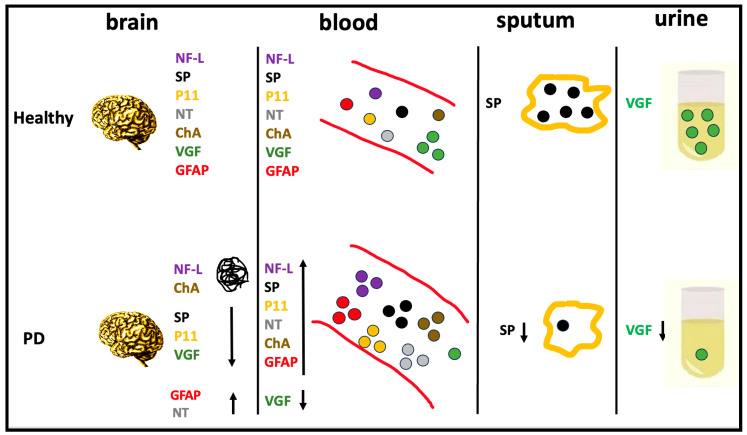
Brain-biomarker changes in bodily fluids of PD patients. Figure legend: The figure shows proteins that change in the brain (nigrostriatal areas) and bodily fluids (plasma, saliva, and urine) of PD patients. NF-L and ChA were found to be abnormally aggregated in the brain and increased in the blood. SP and P11 decreased in the brain (and SP also decreased in sputum), but increased in the blood, while GFAP and NT increased in both the brain and blood. In contrast, VGF decreased in the brain, but also in blood and urine. NF-L: neurofilament light protein; SP: substance P; P11: S100A10 (S100 calcium-binding protein A10); NT: neurotensin; ChA: Chromogranin A; VGF: no acronym; GFAP: glial fibrillary acidic protein. Upward arrow: increase; downward arrow: decrease.

**Table 1 ijms-24-10932-t001:** Changes in protein biomarkers in brains and bodily fluids of PD patients.

Protein		Brain				Body Fluid		
	Method	Change	Sample	Phase	Method	Change	Sample	Phase
NF-L	IHC [17]	*	SN	Advanced Early	SIMOA [21,22,23,24] ECLIA [25]	↑	Serum	Advanced Prodromic
					ECLIA [20]	↑	Plasma	Advanced
SP	RIA [44,45,46]	↓	SN	Advanced	ELISA [50]	↑	Serum	Advanced
	RIA [45,47]	↓	Putamen	Advanced	ELISA [49]	↓	Sputum	Early Advanced
	RIA [48]	↓	Caudate	Advanced				
	RIA [46]	↓	GP	Advanced				
P11	qPCR [53]	↓	SN	Advanced	Flow cytometry [53]	↑	Plasma	Advanced
NT	HPLCRIA [58,59]	↑ ↓	SN Hyp.	Advanced	RIA [60]	↑	Plasma	Early Advanced
ChA	IHC [66,67]	*	SN cerebellum cerebrum	Advanced	ELISA [68]	↑	Serum	Early Intermediate Advanced
VGF	ELISA [87,88,89]	↓	Cortex	Advanced	ELISA [87]	↓	Plasma	Early Advanced
					LC-MS/MS [91]	↓	Urine	with or without LRRK2 G2019S mutation
GFAP	IHC [101]	↑	SN	Advanced	SIMOA [102]	↑	Plasma	Advanced
					ELISA [103] [104] SIMOA [106]	↑	Serum	Advanced

PD = Parkinson’s disease; NFL = neurofilament light chain; IHC = immunochemistry; * = abnormal aggregation; SN = substantia nigra; SIMOA = single-molecule array; ECLIA = electrochemiluminescence immunoassay; SP = substance P; RIA = radioimmunoassay; ELISA = enzyme-linked immunosorbent assay; GP = globus pallidus; P11 = S100A10 (S100 calcium-binding protein A10); qPCR = quantitative polymerase chain reaction; NT = neurotensin; HPLCRIA = high-performance liquid chromatography with radioimmunoassay; ChA = chromogranin A; SgII = secretogranin II; VGF = non-acronym; Ct = C-terminus; LC/MS/MS = liquid chromatography/mass spectrometry/mass spectrometry; LRRK2 = leucine-rich repeat kinase2; G2019S = Glycine2019Serine mutation; GFAP = glial fibrillary acidic protein. Hyp: hypothalamus. ↑ = increased concentration; ↓ = decreased concentration.

## Data Availability

This is a review, the authors reported other published studies, hence they have not the permission to share the results.

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
