# Peer review of "Brain-Biomarker Changes in Body Fluids of Patients with Parkinson’s Disease"

_ijms, 2023, doi:10.3390/ijms241310932_

Round 1

Reviewer 1 Report

In this Review Authors have described the need for Parkinson's disease biomarker for early diagnosis of the disease. Authors mention that Different proteins produced and/or secreted in the substantia nigra are found changed in PD. A good strategy to reveal PD biomarkers is to identify those proteins changing in the brain but also in the body fluids, such as saliva, urine and blood as well as the cerebrospinal fluid (CSF).

Authors have mentioned 7 proteins groups or derived peptides in detail showing studies where they have been shown as significantly regulated in PD brain or biofluids like plasma.

Given the importance of finding the early detecting biomarker, the review gives a good overall picture of the specific proteins mentioned in the Review and how they have been correlated in PD pathogenesis. However, it is very surprising to see Author's have not mentioned about Alpha-Synuclein which has been described in numerous studies as potential PD biomarker. Author's need to explain the rationale of not including this important protein in this review.

Also with so little success in finding a reliable non-invasive early disease biomarker and numerous studies proposing potential non-invasive PD biomarkers, it is very important to include and highlight generic overview of other major proposed markers in the introduction like metabolites, lipids, general proteins. Then Author's can focus on Specific proteins mentioned in this Review, rather then directly describing these proteins. This will make introduction more elaborate and highlight overall progress in the biomarker field.

e.g: Hamid, Z., Basit, A., Pontis, S. et al. Gender specific decrease of a set of circulating N-acylphosphatidyl ethanolamines (NAPEs) in the plasma of Parkinson’s disease patients. Metabolomics 15, 74 (2019). https://doi.org/10.1007/s11306-019-1536-z

Another important point missing in this Review is a representative introductory figure which can make the content of this review more reader friendly and easy to follow. 

Author's need to address the above mentioned comments before i make editorial recomendation of this Article.

English language is fine. Few minor corrections needed.

Author Response

In this Review Authors have described the need for Parkinson's disease biomarker for early diagnosis of the disease. Authors mention that Different proteins produced and/or secreted in the substantia nigra are found changed in PD. A good strategy to reveal PD biomarkers is to identify those proteins changing in the brain but also in the body fluids, such as saliva, urine and blood as well as the cerebrospinal fluid (CSF).

Authors have mentioned 7 proteins groups or derived peptides in detail showing studies where they have been shown as significantly regulated in PD brain or biofluids like plasma. Given the importance of finding the early detecting biomarker, the review gives a good overall picture of the specific proteins mentioned in the Review and how they have been correlated in PD pathogenesis. However, it is very surprising to see Author's have not mentioned about Alpha-Synuclein which has been described in numerous studies as potential PD biomarker. Author's need to explain the rationale of not including this important protein in this review.

 Following the identification of the critical role of alpha-synuclein in PD brain pathology, a large number of studies have summarized the changes in this protein in body fluids, particularly CSF. However, methodological problems have led to a poor diagnostic/prognostic value of alpha-sin and alternative biomarkers are being investigated. Therefore, the authors decided to focus exclusively on protein biomarkers alternatives to alpha-syn, indeed, to the author's knowledge, reviews focusing exclusively on these types of biomarkers are rare. The authors have discussed this point in the new version of the manuscript (paragraph 1.2 entitled: PD Biomarkers: general information). All changes in the paragraph are highlighted in red.

Also with so little success in finding a reliable non-invasive early disease biomarker and numerous studies proposing potential non-invasive PD biomarkers, it is very important to include and highlight generic overview of other major proposed markers in the introduction like metabolites, lipids, general proteins. Then Author's can focus on Specific proteins mentioned in this Review, rather then directly describing these proteins. This will make introduction more elaborate and highlight overall progress in the biomarker field. e.g: Hamid, Z., Basit, A., Pontis, S. et al. Gender specific decrease of a set of circulating N-acylphosphatidyl ethanolamines (NAPEs) in the plasma of Parkinson’s disease patients. Metabolomics 15, 74 (2019). https://doi.org/10.1007/s11306-019-1536-z

The authors have extensively revised the introduction, and a new paragraph (1.2 entitled: PD Biomarkers: general information) has been added. All changes are highlighted in red.

Another important point missing in this Review is a representative introductory figure which can make the content of this review more reader friendly and easy to follow.  Author's need to address the above mentioned comments before i make editorial recomendation of this Article.

Answer: The new version of the manuscript contains a figure (please find the Fig. 1 in the new version of the manuscript) that authors believe is representative of the main content of the review.

Reviewer 2 Report

I have reviewed the manuscript entitled by, “Proteins changing in body fluids and brain of Parkinson’s disease”. The manuscript reviewed and summarized the proteins changing in PD brain as well as in body fluids.

There are a few suggestions.

Major comment:

1.     Highlight the novelty of this review article.

2.     Regarding the title of this manuscript, it’s too raw, please change this title.

Minor comment:

Check the whole manuscript for any typographical and grammar errors.

Author Response

I have reviewed the manuscript entitled by, “Proteins changing in body fluids and brain of Parkinson’s disease”. The manuscript reviewed and summarized the proteins changing in PD brain as well as in body fluids.

There are a few suggestions.

Major comment:

Highlight the novelty of this review article.

The aim of this review is to focus exclusively on protein biomarkers alternatives to alpha-syn, in particular those whose changes in biofluids, (mainly blood, saliva or urine), are associated with nigrostriatal changes. To the best of author’s knowledge, reviews focusing exclusively on these types of biomarkers are rare, and this represents the novelty of the review article. The authors discussed this point in the new version of the manuscript (within the new paragraph 1.2 entitled: PD Biomarkers: general information). All changes are highlighted in red.

Regarding the title of this manuscript, it’s too raw, please change this title.

Title has been changed into:      Brain-biomarker changes in body fluids of patients with Parkinson’s disease

Minor comments:

Check the whole manuscript for any typographical and grammar errors.

We carried out a general revision of the manuscript to check for English grammar, spelling and typing errors.

Reviewer 3 Report

In the current review, Cocco et al. summarized a collection of biochemical changes in the brain and/or in the body fluid (particularly in the blood) in Parkinson’s disease that could serve as biomarkers to aid early detection and differential diagnosis of Parkinson disease.

Major comments:

1.       Authors only covered a couple proteins/peptides in the current review. Additional more promising PD biomarkers, such as α-synuclein, should also be discussed here.

2.       Authors may also include recent progress from two major PD biomarker initiatives: Parkinson’s Disease Biomarkers Program (PDBP) and Parkinson’s Progression Markers Initiative (PPMI).

3.       The proteins/peptides that authors covered in the current review may not exclusively change in PD. For examples, NfL was also shown to be increased in many other neurodegenerative diseases, sometimes higher than PD samples. Authors should discuss how these proteins/peptides could serve as biomarkers for differential diagnosis of PD.

Minor comments:

1.       It is not accurate to state that blood tests “represent the gold-standard in the clinical practice”. Authors should revise the statement.

Minor editing of English language may help further enhance the overall quality of the article.

Author Response

body fluid (particularly in the blood) in Parkinson’s disease that could serve as biomarkers to aid early detection and differential diagnosis of Parkinson disease.

comments: 

Authors only covered a couple proteins/peptides in the current review. Additional more promising PD biomarkers, such as α-synuclein, should also be discussed here.

Following the identification of the critical role of alpha-synuclein in PD brain pathology, a large number of studies have summarized the changes in this protein in body fluids, particularly CSF. However, methodological problems have led to a poor diagnostic/prognostic value of alpha-sin and alternative biomarkers are being investigated. Therefore, the authors decided to focus exclusively on protein biomarkers alternatives to alpha-syn, indeed, to the author's knowledge, reviews focusing exclusively on these types of biomarkers are rare. The authors have discussed this point in the new version of the manuscript (paragraph 1.2 entitled: PD Biomarkers: general information). All changes in the paragraph are highlighted in red.

      Authors may also include recent progress from two major PD biomarker initiatives: Parkinson’s Disease Biomarkers Program (PDBP) and Parkinson’s Progression Markers Initiative (PPMI).

Some results from the PDBP and PMB initiatives were already included in the old version, but they were described without specifying that they came from these initiatives. In the new version of the manuscript, the results of these initiatives have been improved and the old ones have been made more detailed. All changes related to this point are highlighted in yellow.

The proteins/peptides that authors covered in the current review may not exclusively change in PD. For examples, NfL was also shown to be increased in many other neurodegenerative diseases, sometimes higher than PD samples. Authors should discuss how these proteins/peptides could serve as biomarkers for differential diagnosis of PD.

The reviewer is right, the diagnostic accuracy of serum NF-L is indeed low because its changes are not specific for PD. Therefore, rather than being a diagnostic tool for PD, NF-L may be a useful marker for certain specific aspects of PD, such as dementia, cognitive decline or cardiac damage. It may also be a good biomarker for distinguishing PD from ET. In fact, PD biomarkers could also be used to track disease progression or to differentiate PD from other diseases, since the research goal is not just diagnosis. The authors have discussed this point in the new version of the manuscript. All changes related to this point are highlighted in red. Furthermore, in the conclusion, the authors have stated that the limitation of protein biomarkers might be the low specificity, which is the case for NF-L, GFAP or chromogranins. In general (as discussed in the conclusion section of the new manuscript), further studies are needed to clarify whether these proteins could increase their diagnostic or prognostic (or other) value when measured with other fluid biomarkers.

Minor comments:

       It is not accurate to state that blood tests “represent the gold-standard in the clinical practice”. Authors should revise the statement.

 We have deleted it

Comments on the Quality of English Language

Minor editing of English language may help further enhance the overall quality of the article.

We carried out a general revision of the manuscript to check for English grammar, spelling and typing errors.

Round 2

Reviewer 1 Report

Authors have significantly improved the Review and answered all my comments. I recommend this for publication in current form.

minor editing required

Author Response

We have revised the manuscript to correct any typographical or other errors.

Corrections are in red.

We are grateful to the reviewer for his help in revising the manuscript.

Reviewer 2 Report

The revised manuscript seems good and acceptable.

The changed title looks more relevant now.

Good work by authors.

Author Response

(The authors gave the same response as above.)

Reviewer 3 Report

The manuscript was significantly improved in this revision with the additional figure. The authors were able to address most of the comments. 

Minor comment: alpha-synuclein was sometimes named as “α-synuclein” or “alpha-syn” in the text. The authors should be consistent with the protein name.

N/A

Author Response

We have revised the manuscript to correct alpha-synuclein but also any typographical or other errors.

Corrections are in red.

We are grateful to the reviewer for his help in revising the manuscript.